

**Novel aerosol extinction coefficients and lidar ratios over the ocean from**
**CALIPSO-CloudSat: Evaluation and global statistics**
David Painemal[1,2], Marian Clayton[1,2], Richard Ferrare[2], Sharon Burton[2], Damien Josset[3], and
Mark Vaughan[2]
[1]Science Systems and Applications Inc., Hampton, VA, 23666 USA
[2]NASA Langley Research Center, Hampton, VA, 23666 USA
[3]US Naval Research Laboratory Stennis Space Center, MS, 39529, USA
*Correspondence to:* David Painemal (david.painemal@nasa.gov)
**Abstract.** Aerosol extinction coefficients ($\sigma_a$) and lidar ratios (LR) are retrieved over the ocean
from CALIOP attenuated backscatter profiles by solving the lidar equation constrained with
aerosol optical depths (AOD) derived by applying the Synergized Optical Depth of Aerosols
(SODA) algorithm to ocean surface returns measured by CALIOP and CloudSat's Cloud Profiling
Radar. $\sigma_a$ and LR are retrieved for two independent scenarios that require somewhat different
assumptions: a) a single homogeneous atmospheric layer (1L) for which the LR is constant with
height, and b) a vertically homogeneous layer with a constant LR overlying a marine boundary
layer with a homogenous LR fixed at 25 sr (2-layer method, 2L).  These new retrievals differ from
the standard CALIPSO version 4.1 (V4) product, as the CALIOP-SODA method does not rely on
an aerosol classification scheme to select LR. CALIOP-SODA $\sigma_a$ and LR are evaluated using
airborne high spectral resolution lidar (HSRL) observations over the northwest Atlantic. CALIOP-
SODA LR (1L and 2L) positively correlates with its HSRL counterpart (linear correlation
coefficient $r > 0.67$), with a negative bias smaller than 13.2%, and a good agreement for $\sigma_a$ ($r$
$\geq 0.78$) with a small negative bias ($\leq |-9.2 \%|$). Furthermore, a global comparison of optical depths
derived by CALIOP SODA and CALIPSO V4 reveals substantial differences over regions
dominated by dust and smoke, in qualitative agreement with previously reported discrepancies
between MODIS and CALIPSO AOD.
Global maps of CALIOP-SODA LR feature high values over littoral zones, consistent with
expectations of continental aerosol transport offshore. In addition, seasonal transitions associated



with biomass burning during June to October over the southeast Atlantic are well reproduced by
CALIOP-SODA LR.
**1. Introduction**
Advances in our understanding of the 3D structure of atmospheric aerosols have been greatly
accelerated with the advent of the Cloud-Aerosol Lidar with Orthogonal Polarization (CALIOP),
onboard the Cloud-Aerosol Lidar and Infrared Pathfinder Satellite Observation (CALIPSO,
Winker et al., 2009; 2010, 2013). CALIOP has provided the first global view of aerosol distribution
in the boundary layer and free troposphere (Winker et al., 2013), progressed our knowledge of the
long-range transport of dust (e.g. Liu et al., 2008; Uno et al., 2010; Yu et al., 2015) and smoke
(e.g. de Laat et al., 2012; Das et al., 2017; Khaykin et al., 2018), and facilitated the evaluation of
chemical transport models (Nowottnick et al., 2015; Koffi et al., 2016), among many other
accomplishments in the area of aerosol and cloud research.
CALIOP estimates aerosol extinction coefficients on a global scale with an unprecedented
vertical detail. The undetermined problem of solving the lidar equation with two physical
unknowns, the aerosol extinction and backscatter coefficients, is addressed in the CALIPSO
algorithm by relating both variables via an extinction-to-backscatter ratio, or lidar ratio (LR). This
standard technique (e.g. Fernald, 1984) expresses the lidar equation in terms of only one unknown,
if LR is prescribed.  As aerosol types can be related to specific values of lidar ratios (e.g. Müller
et al., 2007), the CALIPSO algorithm utilizes predefined LR assigned to a number of aerosol types,
which in turn, are identified using the CALIPSO automated aerosol typing algorithm (Omar et al.,
2009; Kim et al., 2018). Thus, the quality of CALIOP retrievals will depend on how well the actual
lidar ratios match the pre-tabulated values and to what extent the aerosol typing algorithm properly
classifies aerosols. Another source of uncertainty is the detectability limits of the CALIPSO
algorithm, which prevents retrieving aerosol properties for tenuous aerosol layers (Rogers et al.,
2014; Thorsen et al., 2017). For instance, Toth et al. (2018) found that no aerosol was detected
within ~71% of the CALIOP profiles measured during daytime and ~41% of the nighttime
measurements.  More aerosol detection during nighttime is explained by the absence of solar
background noise, which leads to a significantly better signal to noise ratio. The aforementioned
factors likely explain discrepancies between CALIOP and other remote sensing datasets such as



those from the MOderate resolution Imaging Spectroradiometer (MODIS) and AERONET (e.g.
Redemann et al., 2012; Schuster et al., 2012).
Uncertainty reduction in the selection of LR can be attained by constraining the lidar
equation solution with an independent estimate of aerosol optical depth (AOD). This implies the
minimization of the error between the retrieved AOD (estimated from the retrieved extinction
coefficient coefficient) and the target AOD by iteratively adjusting LR. Burton et al. (2010) utilize
AOD from the MODIS instruments on board both Aqua and Terra satellites for estimating aerosol
extinction from CALIOP for cases in which AOD exceeds 0.15 (0.2) over the ocean (land).
Similarly, Royer et al. (2010) applied an equivalent method for estimating LR and extinction
coefficients over the Po Valley in Italy. Although CALIOP-MODIS retrievals in Burton et al.
(2010) tend to compare better with airborne measurements relative to CALIPSO standard product
(Version 2), MODIS AOD is limited to daytime, and MODIS and CALIOP differ in their along-
track spatial resolution. These previous studies have proven the value of counting on independent
CALIOP retrievals for evaluating CALIPSO's standard data products.
In this contribution, we present a new method in which CALIOP-based lidar ratios and
aerosol extinction coefficients over the non-polar oceans are obtained by constraining the retrievals
with AOD derived from cross-calibrated CALIOP and CloudSat Cloud Profiling Radar (CPR)
surface echos, using the Synergized Optical Depth of Aerosols (SODA) product (Josset et al.,
2008). SODA AOD is a suitable dataset, as it is collocated with CALIOP by definition and
retrievals are possible during both daytime and nighttime for the period 2006-2011. After
November 2011 SODA is only available for daytime, as CloudSat has operated in daylight-only
operations mode to conserve power (Gravseth and Piepe, 2013). Our goal is to provide an
independent CALIOP dataset that can be used for evaluating specific aspects of the CALIPSO
Science Team product, as well as for investigating aerosol-related topics in climate research. We
first summarize the algorithm and evaluate the new retrievals against state-of-the-art aerosol
observations from the NASA Langley airborne High Spectral Resolution Lidar-1 (HSRL, Sections
3 and 4). Next, we compare the CALIOP-SODA extinction coefficient and AOD with their
CALIPSO Science Team Version 4 counterparts. Lastly, we present global maps of lidar ratio and
marine boundary layer aerosol optical depth, and provide a physical interpretation for the regional
patterns derived from CALIOP-SODA.





## 2. Dataset

### 2.1. CALIOP

Version 4.1 (V4) CALIOP elastic backscatter lidar measurements at 532 nm and 1064 nm are utilized in this work. For the derivation of CALIOP-SODA retrievals, we use Level 1 lidar attenuated backscatter and the Level 2 Vertical Feature Mask product, with a 333 m horizontal resolution below 8.2 km. CALIOP V4 aerosol extinction coefficients and AOD estimates are taken from the Level 2 Aerosol Profile product at 5 km horizontal resolution. For comparing CALIOP SODA and V4 products, we follow the procedure outlined in Koffi et al. (2016): where the VFM feature classification flags indicate regions of clear air, we set the corresponding extinction coefficients to zero. While these regions are labeled as 'clear air', they are simultaneously assumed to be populated by highly diffuse aerosols that lie well below the CALIOP layer detection threshold.

### 2.2. SODA aerosol optical depth

SODA uses the relationship between CALIOP (532 nm and 1064 nm) and CPR (3.1 mm, 94 GHz) surface return signals, along with a correction for the atmospheric transmission at the radar wavelength, to derive AOD at the lidar wavelengths. In short, SODA estimates of AOD rely on the radar-to-lidar ocean surface scattering cross-calibration for cloud-free columns (Josset et al., 2008, 2010). Consequently, SODA can provide a cloud-free AOD without having to rely on an accurate assignment of a particular aerosol type with an appropriate lidar ratio. In addition, the algorithm does not depend on pre-determined aerosol models with a specific particle size distributions and refractive indexes, unlike MODIS. SODA AOD Version 2, based on CALIPSO Version 3 (V3), is developed at the ICARE data and services center (http://www.icare.univ-lille1.fr) in Lille (France) under the auspices of the CALIPSO mission and supported by the French National Centre for Space Studies (CNES). Josset et al. (2013) estimate a systematic error in SODA AOD of 0.015 and 0.059, respectively, for nighttime and daytime AOD. In addition, good agreement between SODA and MODIS has been reported in Josset et al. (2010, 2015), while Dawson et al. (2015) reports a root-mean-square-error of 0.03 between SODA and AERONET AOD and $r = 0.59$ for AERONET sites near the coast. Further, we also evaluate SODA AOD with HSRL data in Section 4, and compare SODA and MODIS AOD over the global ocean in Section 6. While 1064 nm SODA AOD is also utilized in this study, caution needs to be exercised when using the 1064 nm SODA data due to uncertainty calibrations in CALIPSO V3 (Vaughan et al., 2010).





**2.3. HSRL**
CALIOP retrievals are evaluated against airborne measurements by the NASA Langley
High Spectral Resolution Lidar (HSRL, Hair et al., 2008) at 532 nm. The instrument allows for
the independent determination of aerosol extinction and backscatter coefficients at 532 nm (and
thus, lidar ratio) using the HSRL technique (Eloranta, 2005). As HSRL measurements at 1064 nm
are limited to attenuated backscatter, similar to CALIOP, only 532 nm HSRL retrievals will be
utilized in this study. The data used in this study were acquired August 11–27, 2010 while the
HSRL conducted a dedicated CALIPSO validation campaign over the Caribbean Sea (Burton et
al., 2013; Rogers et al., 2014).  As required for all HSRL-CALIPSO validation measurements, the
HSRL flight paths during this campaign were spatially matched with coincident CALIPSO ground
tracks (Rogers et al., 2014).
**3. Derivation of aerosol extinction coefficient and lidar ratio**
The method for deriving aerosol extinction coefficient ($\sigma_a$) and lidar ratio (LR) is based on
Fernald (1984) applied to the CALIOP attenuated backscatter, and is briefly summarized in the
following. For CALIOP, the lidar equation is expressed in terms of height $z$ (range) as:

$$\beta_{att}(z) = \left(\beta_m(z) + \beta_a(z)\right) \cdot exp\left(-2\int_0^z \left(\sigma_m(z') + \sigma_a(z')\right)dz'\right) \quad (1)$$

Where $\beta_{att}$ corresponds to the CALIOP total attenuated backscattering cross section, $\beta_m$ and $\beta_a$
denote the molecular ($m$) and aerosol ($a$) backscatter coefficients, and $\sigma_m$ and $\sigma_a$ are the molecular
and aerosol extinction coefficients. Since the molecular contribution can be accurately estimated
using atmospheric profiles from numerical weather models, the two unknowns are $\beta_a(z)$ and
$\sigma_a(z)$. Equation (1) can be reduced to one unknown by relating extinction and backscatter
coefficient via their lidar ratio, that is

$$LR = \frac{\sigma_a(z)}{\beta_a(z)}. \quad (2)$$

It follows that eq. (1) can be expressed in terms of LR and $\beta_m$ as:

$$\beta_{att}(Z) = \left(\beta_m(z) + \beta_a(z)\right) \cdot exp\left(-2\int_0^z \left(\sigma_m(z') + LR \cdot \beta_a(z')\right)dz'\right) (3)$$

$\beta_m$ is estimated as a function of air density (from the Goddard Earth Observing System
Model, Version 5 GEOS-5), and effect of ozone attenuation is accounted for in $\sigma_m$ following





Vaughan et al. (2005). $\beta_a(z)$ and $\sigma_m(z)$ are, thus, estimated from Eq. (3) as in Fernald (1984),
assuming a constant value of LR with height. The LR selection is physically constrained by
comparing the retrieved aerosol optical depth ($AOD_{ret} = \int_0^z \sigma_a(z')dz'$) with SODA AOD
($AOD_{SODA}$), and LR is iteratively adjusted until  the retrieved AOD matches the SODA AOD to
within 0.001 or less (i.e., when $|AOD_{ret} - AOD_{SODA}| \leq 0.001$).

6       We also consider an additional scenario for solving the lidar equation, which consists of

treating the atmospheric column as two layers, that is., the marine atmospheric boundary layer
(MBL) and a second aerosol layer of as-yet-undetermined composition. This method is intended
to better capture specific events with two predominant aerosol types, namely, smoke over marine
aerosols and dust over marine aerosols, which are particularly frequent over the Atlantic Ocean.
The LR for the MBL is assumed constant at 25 sr, as suggested by HSRL measurements over the
ocean (Burton et al., 2012; 2013). This lidar ratio is slightly higher than the one compiled by Kim
et al. (2018) for maritime aerosols (23 sr). $\sigma_a(z)$ and the upper layer LR are iteratively calculated
using the Fernald method with the constraint provided by $AOD_{SODA}$, and LR =25 sr in MBL. MBL
height is computed by applying the bulk Richardson number method (McGraw-Spangler and
Molod, 2014) to GEOS-5 atmospheric fields.
In sum, aerosol extinction profiles and lidar ratio are calculated using one of two
independent assumptions: either a single homogeneous layer (1L method), or a two-layer
atmosphere with a prescribed LR=25 sr in the boundary layer and a single free tropospheric LR,
which is estimated during the iterative process (2L method).
The CALIOP attenuated backscatter at 333 m resolution is taken from the Level 1
CALIPSO product. Before retrieving LR and $\sigma_a$, three contiguous 333-m lidar attenuated
backscatter samples are averaged to achieve a 1 km along-track resolution. Similarly, SODA AOD
retrieved at 333m is averaged to 1 km resolution. In addition, the Feature Classification Mask
product is utilized for identifying cloudy pixels and cases with fully attenuated signal, in which
CALIOP-SODA retrievals are not possible.
**4. CALIOP-SODA evaluation with airborne HSRL measurements**
CALIOP-SODA retrievals of aerosol extinction coefficient, lidar ratio and AOD are
evaluated using eight flights during August 2010 over the western Atlantic, for the domain
bounded by 70˚W-55˚W and 13˚N-35˚N (Figure 1a). CALIOP-SODA is spatially averaged to



match the nominal 5 km horizontal resolution of CALIPSO V4, and only samples with 5-km cloud-
free scenes are retained. Both CALIPSO V4 and CALIOP-SODA are then spatially collocated
with the aircraft track (Figure 1) for samples with a temporal mismatch of less than 90 minutes
(Rogers et al., 2014). Lastly, satellite and airborne observations are spatially averaged to a common
0.5 ˚ resolution (in latitude). Approximately 42 and 46 0.5˚-samples were collocated with HSRL
(CALIOP-SODA and CALIOP, respectively).
The HSRL measurements during Caribbean 2010 were characterized by the presence of
dust, dust mixed with maritime aerosols, and continental pollution; the occurrence of pure
maritime aerosols was confined to the boundary layer (Burton et al., 2013). This aerosol typing is
manifested in a lidar ratio of 25 sr below 500 m, and a linear increase with height that reaches
values of 40-45 sr in the free troposphere (Fig. 1b). These measurements also provide justification
for the use of a lidar ratio of 25 sr in the boundary layer for the 2L method. Before evaluating
aerosols extinction coefficients and lidar ratios, we compare SODA AODs and CALIOP V4 AODs
against their HSRL counterparts (Fig. 2a).  In general, both CALIOP-based retrievals correlate
well with the HSRL, with a slightly higher correlation for SODA, and absolute bias between 10-
17%), with SODA (CALIOP V4) underestimating (overestimating) AOD. Linear fits of SODA
and V4 AOD relative to HSRL (red and blue lines in Fig. 2a) indicate that the SODA bias is
relatively constant with AOD whereas a V4 AOD overestimate tends with increase with AOD
especially during nighttime. Nighttime and daytime correlations remain approximately the same
for both CALIOP V4 and SODA. However, V4 linear correlation coefficient for AOD < 0.3 are
slightly lower for daytime ($r = 0.78$) than nighttime ($r = 0.94$), whereas SODA daytime/nighttime
correlations for low AOD remain high ($r \geq 0.93$). The reduced daytime correlation for CALIOP
V4 is expected as the reduced signal to noise ratio due to the solar background signal hampers the
algorithm's ability to detect and classify aerosols. Finally, in terms of the root-mean-square error
(RMSE), SODA RMSE (24.2% relative to the mean) is smaller than that for CALIOP V4 (31.2%,
Table 1).
The evaluation of CALIOP-SODA lidar ratio and aerosol extinction coefficient is
summarized in the following. For LR, we use the column-effective lidar ratio, calculated as:

$$LR_{HSRL} = \frac{\sum_{z=z0}^{6km} \sigma_a(z)}{\sum_{z=z0}^{6km} \beta_a(z)} \qquad (4)$$



For evaluating CALIOP SODA 1L LR, $LR_{HSRL}$ in eq. (4) is estimated using the last range bin above
the ocean surface (37.8 m) as the lower bound, $z_0$. In addition, the comparison between CALIPSO
SODA 2L LR and $LR_{HSRL}$ is performed by recomputing $LR_{HSRL}$ using the MBL height for $z_0$ in eq.
4. Since valid HSRL extinctions retrievals are only derived for heights above 270 m from the
surface, we have assumed a constant extinction coefficient for the layer below 270 m, with values
taken from the lowest height with available retrievals (~ 270 m). The comparison depicted in Fig.
2b, yields $r$ = 0.67-0.74 between both CALIOP-SODA methods (1L and 2L) and HSRL, with a
negative mean bias smaller than 13%, and RMSE of up to 8.1 sr (Figure 2b and Table 2).

9        Mean vertically-resolved aerosol extinction coefficients from SODA, CALIOP V4, and

HSRL are depicted in Figure 3a and b for daytime and nighttime observations, respectively. The
agreement between HSRL (red) and CALIOP-SODA 1L and 2L (overlapped gray and black) is
remarkable throughout the lower troposphere, with a maximum overestimation of 0.027 km$^{-1}$
(50%) near 500 m. CALIOP-SODA 1L and 2L yield identical results, which is likely the effect of
a shallow marine boundary layer (<500 m). In contrast, CALIOP V4 (blue) consistently
overestimates the airborne measurements for heights below 1 km during both daytime and
nighttime, with magnitudes up to 0.102 km$^{-1}$ (100%) relative to the HSRL during nighttime and
0.078 km$^{-1}$ (140%) during the day. This overestimate is explained by the CALIPSO V4 constant
lidar ratio of 37 sr for dusty marine aerosol, which is generally higher than the lidar ratio retrieved
by both the HSRL and SODA for Caribbean 2010 (Figure 2b). Interestingly, both CALIOP-SODA
and CALIOP V4 correlate well with the HSRL, with correlations around 0.80 (Table 3). The
RMSE for CALIOP V4 is also higher than that for CALIOP-SODA especially below 1 km, with
maxima around 0.12 km$^{-1}$ (155%) and 0.06 km$^{-1}$ (83%) for CALIOP V4 and CALIOP-SODA,
respectively (Fig 3c). Aerosol extinction coefficient statistics for the atmospheric column below
3.0 km (Table 2) corroborate the overall smaller bias and RSME of CALIOP-SODA relative to
V4.
**5. Global analysis**
**5.1. Comparison between CALIOP-SODA and CALIOP-V4**

29        Five months of collocated SODA and CALIOP V4 Level 2 data during June-October 2010

were compared over non-polar oceanic regions with the goal of identifying main differences in
aerosol extinction coefficient profiles. This period was selected because of the high global



climatological AOD observed over the ocean by CALIOP (e.g. Yu et al., 2010). We first averaged
1km CALIOP-SODA to the V4 Level 2 nominal resolution (5km). Then, CALIOP-SODA and
CALIOP V4 retrievals were further reduced by averaging the retrievals to a common 25 km
resolution. Lastly, we reduced the potential effect of overcast scenes in the retrievals by limiting
the comparison to 25-km samples with less than 2/3 (67%) of cloudy coverage. Cloud cover was
derived from the 333 m Vertical Feature Mask and determined as the ratio between profiles with
at least one cloudy feature in the atmospheric column to the total.  To circumvent CALIOP's
narrow field of view, we calculated the statistics in 6˚x3˚ (longitude x latitude) grids.

9         We first focus on the AOD difference (ΔAOD) between CALIOP V4 and SODA at 532

nm and 1064 nm, for day and nighttime (Figure 4). Daytime 532 nm ΔAOD maps reveal higher
V4 AOD than SODA for the northeast Atlantic (NEA) and the Indian Ocean (IO), whereas V4
AOD is smaller than SODA over the southeast Atlantic (SEA) and over vast regions of the open
ocean. As will be discussed in Section 6, these differences are similar to those observed between
CALIOP V3 and MODIS (Redemann et al., 2012). Overall, nighttime differences in 532 nm AOD
appear to diminish especially for the SEA and the northwest Pacific (NWP), while the positive
ΔAOD remains high over IO and NEA.

17         We also show ΔAOD for the 1064 nm channel in Figure 4 (lower panels). The largest

ΔAOD values are mostly confined to the NEA and IO domains, with higher values for SODA
AOD, while nighttime ΔAOD are similar to its daytime counterpart.

20         Matched CALIOP-SODA and CALIOP V4 mean vertical profiles of aerosol extinctions

over the regions defined in Figure 4 (black boxes) are shown in Figs. 5 and 6, for the 532 nm and
1064 nm channels, respectively. CALIOP-SODA derived using the 1L (CAL$_S$ 1L) and 2L (CAL$_S$
2L) methods are depicted in black and blue, while CALIPSO V4 (CAL$_{V4}$) is in red. The main
differences, in agreement in AOD differences in Figure 4, are found: a) over IO and NEA where
CALIPSO V4 extinction profiles are higher than CALIOP-SODA, and b) over SEA, with lower
V4 extinctions than CALIOP-SODA. Even though the main V4-SODA differences in extinction
decrease during nighttime, especially over the SEA, the nighttime differences for NEA and IO
remain nearly the same. Interestingly, the higher CALIOP V4 extinction for NEA and IO
resembles the CALIPSO V4 overestimation during Caribbean 2010 (Fig. 3). CALIOP-SODA and
V4 profiles differences for regions with small AOD differences, such as the south Pacific (SP) and
the northwest Pacific (NWP), are modest. Another interesting aspect is the generally higher



variability of daytime CALIPSO V4 relative to SODA, manifested in the high standard deviations
in Figure 5 (error bars). This indicates that SODA retrievals are more stable than CALIPSO V4
especially during the daytime, due to the AOD constraint. Moreover, the high solar background
substantially degrades CALIPSO aerosol detection capabilities, affecting the retrieved extinction.
Lastly, CALIOP-SODA differences between 1L and 2L are small, and typically confined to a layer
below 700 m, where 2L tends to be smaller than 1L. This is explained, as in Section 4, by a
relatively shallow mixed-layer height ($<$ 500 m), where LR = 25 sr for the 2L method.

8         For completeness, we show in Figure 6 the aerosol extinction profiles for the 1064 nm

channel. CALIOP-SODA and V4 profiles yield smaller differences relative to their 532 nm
counterpart, in agreement with ΔAOD (Figure 4).
**5.2. Maps of CALIOP-SODA Lidar ratio (LR) at 532 nm**

13        Figures 7 and 8 depict global maps of 532 nm LR derived from the 1L ($LR_{1L}$) and 2L

($LR_{2L}$) assumptions, temporally averaged from March to August (MAMJJA, boreal spring-
summer) and September to February (SONDJF, boreal autumn-winter) of 2010 from the 25-km
averaged retrievals with cloud fraction less than 67%. Daytime 532 nm LR exhibits a clear spatial
pattern with high values ($>$45 sr) in coastal regions especially off the southwestern African coast
and east of China. The lowest values are observed over the western and central equatorial Pacific,
with ratios less than 30 sr, which are typical of clean maritime environments (e.g. Burton et al.,
2013). Semiannual transitions are primarily found near the continents, namely, the Southeast
Atlantic, Mediterranean Sea, Indian Ocean, and off the coast of eastern Asia. Nighttime LRs are
similar to their daytime counterparts, but with slightly higher values and a rather heterogeneous
pattern, likely attributed to the reduced cloud-free sampling at night due to the increased cloud
cover, especially over subtropical regions where stratiform and shallow cumulus clouds are
abundant.

26        Comparing the two layer assumptions, $LR_{2L}$ (Figure 9) is higher than $LR_{1L}$, especially for

lidar ratios $>$ 40 sr.  This result is expected, as the prescribed MBL lidar ratio of 25 sr for 2L tends
to be lower than the lidar ratio for any aerosol type that would be found above the marine boundary
layer, and therefore lower than the column average or column effective lidar ratio. Therefore, to
match the SODA AOD, the lidar ratio above the MBL in the 2L case must be larger than the
column effective value that the 1L case derives. Overall, $LR_{1L}$ and $LR_{2L}$ differences are relatively



small (~ 5 sr), which, as we will show in the next section, is associated with the shallow MBL
height estimated from the bulk Richardson number method, and therefore a relatively small
fraction of aerosol that is controlled by the assumed marine lidar ratio in the 2L method.

**5.3. Fractional CALIOP-SODA AOD at 532 nm in the marine boundary layer**

CALIOP-SODA aerosol extinctions are further utilized for quantifying AOD in the
boundary layer. We first show in Figure 9 the 2010 semiannual total SODA AOD for daytime
(left) and nighttime (right) CALIPSO overpasses. Consistent with several studies (e.g. Kittaka et
al., 2011; Redemann et al., 2012), high AOD primarily occur over the eastern Atlantic, in
connection with biomass burning and dust emissions from southern and equatorial Africa. A
second region of interest encompasses most of the Asian coastal region, where a combination of
pollution and dust give rise to high AOD (Itahashi et al., 2010).
Before presenting MBL AOD, we show the MBL height maps (Figure 10), with typical
heights below 800 m, and littoral maxima up to 1150 m in northern Africa and Eurasia. Next, we
compute MBL AOD by numerically integrating CALIOP-SODA aerosol extinction coefficient
from the surface to the MBL height.  MBL AOD in Figure 11 shows a dissimilar pattern relative
to its total AOD counterpart (Figure 9), manifested in a less dominant role of the southeast Atlantic.
In addition, coastal Africa, Eurasia, and North America exhibit peaks in MBL AOD (>0.12) during
boreal spring-summer. A second region with high AOD encompasses the extratropical oceans
poleward of 45 ˚S/N, with a particularly consistent zonal band with high AOD in the Southern
Ocean. As expected, 2L MBL AOD is lower than 1L due to the 2L assumption of a lidar ratio =
25 sr in the MBL. Except for the subtropical ocean, which features shallow MBL and low MBL
AOD, a spatial modulation of the marine boundary layer in the MBL AOD is unclear. It is
important to mention that estimates of the AOD apportioned in the boundary layer will depend on
the MBL dataset utilized in the calculations. An alternative MBL height estimation derived from
CALIOP attenuated backscatter (McGrath-Sprangler and Denning, 2013) yields similar if not
slightly higher values than our GEOS-based MBL. However, MBL estimates based on
thermodynamical vertical profiles (temperature, relative humidty) from meteorological analyses
produce significantly higher MBL (von Engeln and Teixeira, 2013), closely matching the cloud
top height of stratiform and shallow cumulus clouds. Thus, the MBL used here is expected to
primarily represents the mixed-layer height (von Engeln and Teixeira, 2013).



The fraction of MBL AOD relative to the total is depicted in Fig. 12. The extratropical
bands show the highest fraction of MBL AOD, accounting for up to 0.73 (73%) of the total AOD.
Low fractions are found in the subtropics and tropics, with the lowest AOD fraction over the
eastern Atlantic and the west-central Pacific. Interestingly, vast areas over the ocean feature AOD
fractions of less than 40%, suggesting a significant contribution of free tropospheric aerosols to
the total AOD. These results are qualitatively consistent with the results Bourgeois et al. (2018)
using CALIPSO version 4.1.
**6. Discussion**
Due to the limited airborne and ground-based observations of lidar ratios over the ocean, a
global assessment of CALIOP-SODA retrievals is challenging. One of the few global satellite-
based estimates of lidar ratio is reported in Bréon (2013) who estimated LR utilizing the retrieved
scattering phase function at 180˚ angle derived from the Polarization and Directionality of the
Earth's Reflectances (POLDER) satellite instrument and a prescribed aerosol model. POLDER
LR is somewhat comparable to CALIOP-SODA (Figure 8-9) with both retrievals yielding high
LR over the coasts of eastern Africa and Eurasia, and a notable increase in LR over the Indian
Ocean in boreal autumn-winter. In addition, both POLDER and CALIOP-SODA produce LR < 30
sr over the open ocean. On the other hand, LR from POLDER tend to be slightly higher, with a
typical range between 30-70 sr. Bréon (2013) also indicates that because POLDER retrievals rely
on scattered photon measurements, LR might be biased low in regions dominated by absorbing
aerosol, such as the southeast Atlantic. A somewhat different method of retrieving LR from SODA
AOD, documented in Josset et al. (2011), consists of analytically solving the lidar equation. The
only available global analysis of LR using the technique in Josset et al. (2011) is documented in
Dawson et al. (2015) for maritime aerosols only, reporting values between 20-40 sr.
As different aerosol types can be, to some extent, characterized by their lidar ratio, the
reliability of CALIOP-SODA LR retrievals is qualitatively assessed by analyzing the consistency
between the CALIOP-SODA LR spatial pattern and the regional occurrence of aerosol types.
Burton et al. (2012), using HSRL measurements over North America and the adjacent Atlantic
Ocean, provide the following lidar ratios for a number of aerosol types: the highest LR (45-80 sr)
are typically attributed to smoke and urban aerosols, LR of 25-50 sr and 40 sr are associated with
dust and polluted maritime aerosols (respectively), and maritime aerosols are characterized by lidar





ratios of less than 30 sr. Following the classification of Burton et al. (2012), CALIOP SODA lidar
ratios (Fig. 7-8) will be interpreted in the context of major variations in atmospheric aerosols over
the global oceans. For simplicity, we will primarily interpret daytime $LR_{1L}$ in Figures 7a and c.
Two distinct maxima in CALIOP-SODA LR (LR > 50 sr) for spring-summer (Figure 8a) are found
over the southeast Atlantic and the Mediterranean Sea. The LR peak in the southeast Atlantic is
explained by the well-documented biomass burning season over southern Africa, with massive
fires events from May to September during the dry season (Roberts et al., 2009), and smoke being
transported offshore by the prevailing winds during July to October (Adebiyi et al, 2015). The high
spring-summer LR over the Mediterranean Sea is also expected given the southward pollution
transport from Europe which is maximized in summer (Duncan and Bey, 2004). A major LR
maximum in autumn-winter is observed south of India, over the Bay of Bengal and part of the
Arabian Sea. This pattern is concomitant with the pervasive presence of pollution and biomass
burning during the winter and pre-monsoon season (October to April, Krishnamurti et al., 2009).
In contrast, during the monsoon season (June-September), dust aerosols become the dominant
species over the Bay of Bengal (Das et al., 2013), which is manifested in the substantial reduction
in SODA LR in spring-summer. Regions with intermediate CALIOP-SODA LR (35 sr< LR< 50
sr) are located over broad regions of the eastern Pacific, as well as narrow littoral bands over
eastern Asian and the east coast of North America. These regions are likely influenced by a
combination of maritime aerosols and pollution from the continents. Lastly, the regions with the
lowest LR are located over the tropical ocean, where AOD is the lowest (Figure 10). LR over the
Southern Ocean is typically around 30-35 sr, which is rather surprising for such a remote region
with limited continental influence, where maritime aerosols (and associated lidar ratios near 25 sr)
are expected to be the dominant aerosol type. Interestingly, Omar et al. (2009) noted a large
number of cases in which the CALIPSO aerosol classification algorithm detected continental clean
aerosol, which in turn, the CALIPSO algorithm assigns a 532 nm lidar ratio of 35 sr (version 3)
and 53 sr (version 4.1). Lastly, the interpretation the 1064 nm CALIOP-SODA is not attempted
here due to the lack of independent measurements and calibration uncertainties associated with the
use of CALIPSO V3 for deriving SODA AOD. A future release of SODA based on CALIPSO V4
will benefit from the improved calibration of V4, which is estimated to be within 3% (Vaughan et
al., 2018).



An aspect that deserves further discussion is the reliability of SODA AOD, as it is essential
for constraining the lidar equation in our study. In this study we find a high linear correlation
between SODA and HSRL AOD (r=0.96), with no clear relationship between SODA biases and
AOD magnitudes, and a SODA-to-HSRL RSME comparable to the one estimated between SODA
and AERONET in Dawson et al. (2015). The differences between SODA and the CALIPSO V4
AOD (Figure 4) also support inferences based on comparisons between MODIS and CALIPSO
Science Team AOD over the ocean (Redemann et al., 2012; Kim et al., 2013). Analogous to our
results, Redemann et al. (2012) found an underestimation of CALIOP Version 3 of MODIS
collection 5 over the southeast Atlantic in July, and overestimation over the Bay of Bengal and the
zonal band near the equatorial Atlantic Ocean. Redemann et al. (2012) and our results both point
to an overestimation of CALIPSO V4 AOD over oceanic regions dominated by dust, and
underestimation in regions dominated by smoke. To verify that SODA-CALIPSO V4 differences
are mainly attributed to CALIPSO V4 biases, we perform an additional comparison using Aqua-
MODIS Level 2 550 nm AOD (MYD04_3K product), taken from the latest Collection 6.1 (Levy
et al., 2013) for the June to September period of 2010. Cloud-free 3-km MODIS AOD pixels are
collocated with CALIPSO track and averaged to approximately 25 km (along track) to match the
averaged 25 km SODA retrievals. Next, MODIS-SODA mean differences are averaged every
6˚x3˚ grid, and depicted in Figure 13. The MODIS-SODA differences in Figure 13 are typically
within the [0.06 -0.06] range, with negligible values over the eastern Atlantic and the northwest
Pacific. Although ΔAOD reaches up to 0.12 over the Indian Ocean, these differences are smaller
than those between CALIPSO V4 and SODA (Figure 4, upper left panel). Overall, MODIS further
corroborates that CALIPSO V4 AOD is biased over regions dominated by smoke and dust. We
note that the plausible oceanic CALIOP V4 bias dependence on aerosol types suggested in our
study might not be applicable over land, where AOD for dust is underestimated by CALIPSO (e.g.
Schuster et al., 2012).
**7. Concluding remarks**
One year of a new CALIOP-based aerosol extinction coefficient and lidar ratio dataset has
been presented, with the goal of providing a flexible dataset for climate research as well as
independent retrievals that can be helpful for refining CALIPSO Science Team algorithms. The
new retrievals build on the CALIPSO V4 total attenuated backscatter and cloud mask data





products. However, the method that we used to invert the lidar equation differs fundamentally from
the CALIOP standard aerosol product, as it does not rely upon an aerosol classification module to
prescribe the lidar ratio. We evaluated CALIOP-SODA AOD, LR, and extinction using airborne
HSRL retrievals over the western Atlantic, and found excellent agreement, with statistically
significant correlations and biases less than 27 %. Given these encouraging results, we envision
potential uses of CALIOP-SODA lidar ratios for evaluating CALIOP V4 aerosol properties. This
can be done similar to Dawson et al. (2015), by stratifying CALIOP-SODA LR as a function of
CALIOP V4 aerosol types and their assigned lidar ratio.
Although the retrievals presented here are limited to cloud-free atmospheric columns due
to the constraint imposed by SODA AOD, it is possible to adapt the algorithm to make use of
above-cloud satellite AOD retrievals (e.g., Jethva et al., 2014; Liu et al., 2015). In this regard,
above-cloud AOD using CALIOP can be derived by combining the integrated attenuated
backscatter and depolarization ratio (Hu et al., 2007; Liu et al., 2015), with corrections for the
multiple-scattering depolarization relationship implemented by SODA (Deaconu et al., 2017).
Efforts to retrieve above-cloud lidar ratio and extinction profiles over the southeast Atlantic using
the above cloud AOD are currently underway (Ferrare et al., 2018).
CALIOP-SODA 1L retrievals are expected to perform better for relatively homogeneous
atmospheric profiles characterized by a single aerosol type. Alternatively, SODA 2L retrievals are
likely to be advantageous for specific regions where massive aerosol plumes from the continent
are transported offshore at high altitudes through convective processes, in such a way that MBL
aerosols are detached from the layer above and the assumption MBL LR=25 sr (maritime) is a
good approximation. This is probably the case over the southeast Atlantic during the biomass
burning season or for episodic dust transport over the tropical Atlantic. However, the CALIPSO
Science Team product will continue providing the best available global dataset for monitoring
complex aerosol profiles, continental processes, and aerosols in the upper troposphere.
*Data availability. CALIPSO version 4.1 is available at https://eosweb.larc.nasa.gov, and SODA*
*aerosol optical depth at* http://www.icare.univ-lille1.fr.
*Competing interests.* The authors declare that they have no conflict of interest.




*Acknowledgements: This work was funded by the CloudSat and CALIPSO Science Recompete*
*Program NASA award # NNH16CY04C.* The SODA product is developed at the AERIS/ICARE
data and services center (http://www.icare.univ-lille1.fr/projects/soda) in Lille (France) in the
frame of the CALIPSO mission and supported by CNES. The Aeris data infrastructure is greatly
acknowledged for data, processing and development supports of the SODA product. We thank Dr.
Gregory Schuster for his insightful comments and suggestions.

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





# 1 Figures

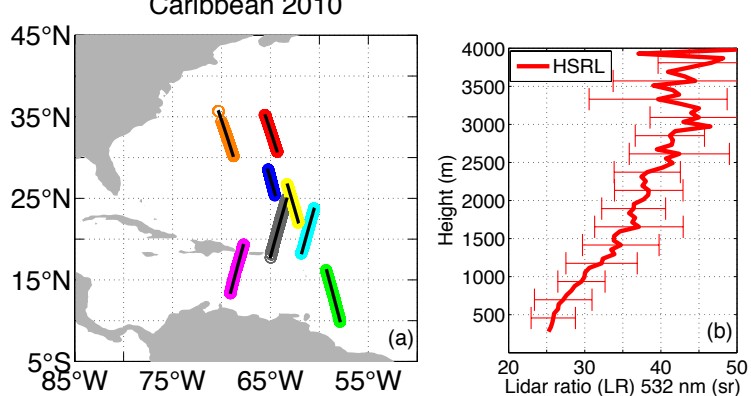

Figure 1: a) Flight tracks during the 2010 field campaign. Black solid lines correspond to the

matched CALIPSO tracks. b) Mean HSRL lidar ratio (532 nm) as a function of altitude and one

standard deviation (error bar) for all the flight tracks in Fig. 1a.





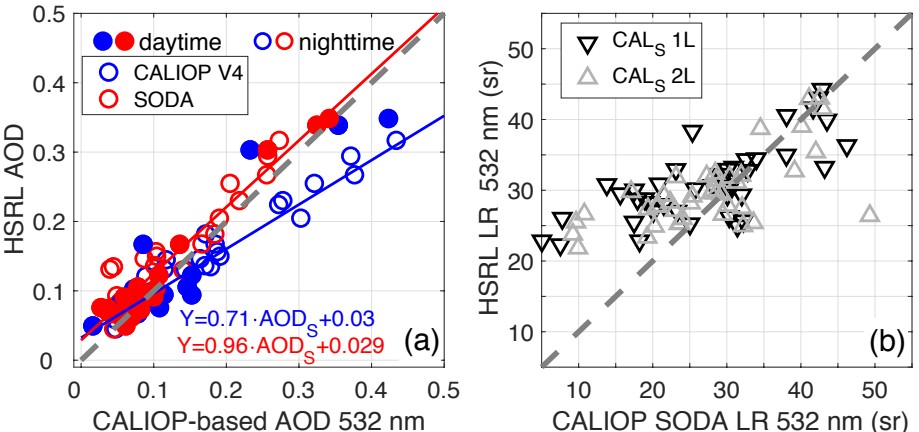

Figure 2: a) Scatterplot between SODA (red) and CALIPSO V4 (blue) against HSRL AOD at
532 nm. Filled and open circles indicate daytime and nighttime observations, respectively. Blue
and red lines (and equations) are the linear fit for V4 and SODA AOD ($AOD_{v4}$ and $AOD_S$)
relative to HSRL. b) Comparison between CALIPSO SODA ($CAL_S$) lidar ratio based on the 1-
layer (1L) and 2-layer (2L) assumption with the HSRL column-effective lidar ratio from Eq. 4
(black and gray symbols, respectively). Gray dashed line is the one-to-one relationship.





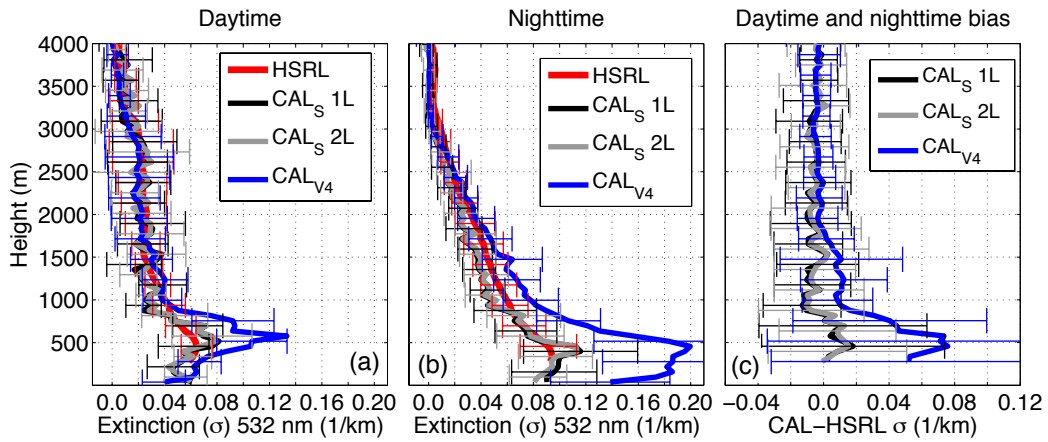

Figure 3: Mean aerosol extinction coefficient profile from the HSRL (red), CALIPSO SODA 1L

4        (black), 2L (gray), and CALIPSO standard V4 product (blue) during a) daytime and b)

nighttime. c) Total mean bias of CALIPSO-based extinction relative to the HSRL: CALIPSO
SODA 1L (black) and 2L (gray), CALIPSO V4 (blue). Error bars in Fig. 3 a and b denote one

7                standard deviation, and RMSE in Fig. 3c.





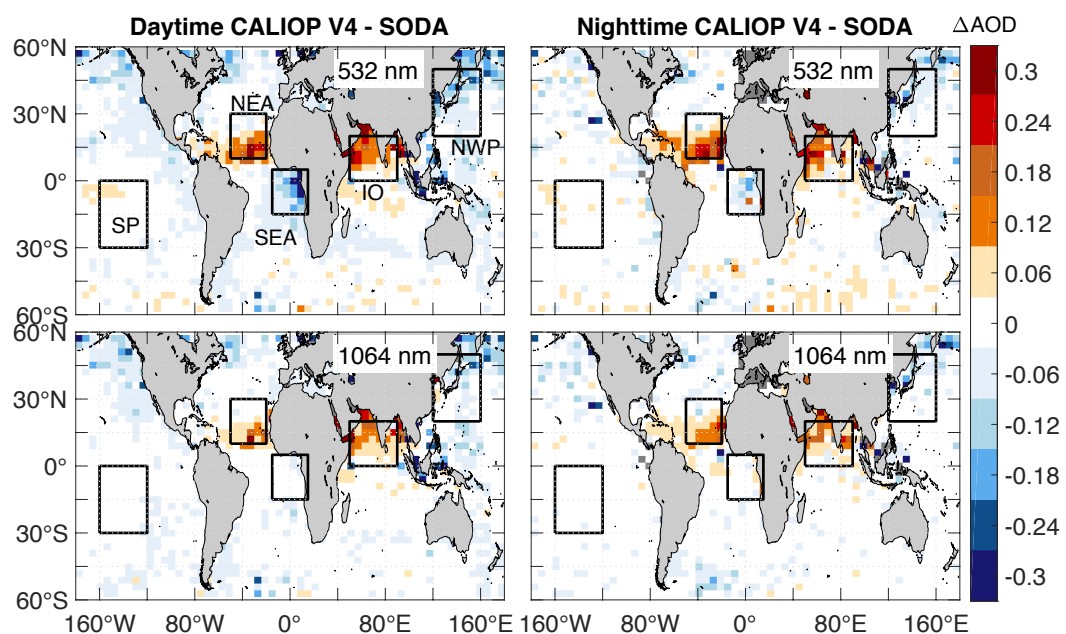

Figure 4: Mean AOD difference between CALIOP V4 and SODA for five months of 2010 for
daytime (left) and nighttime (right), and the 532 nm (upper panels) and 1064 nm (lower panels)
channels. Boxes denote specific regions in which the extinction coefficient profiles are further
compared in Figure 5: South Pacific (SP), southeast Atlantic (SEA), Indian Ocean (IO), northeast
Atlantic (NEA), and northwest Pacific (NWP).






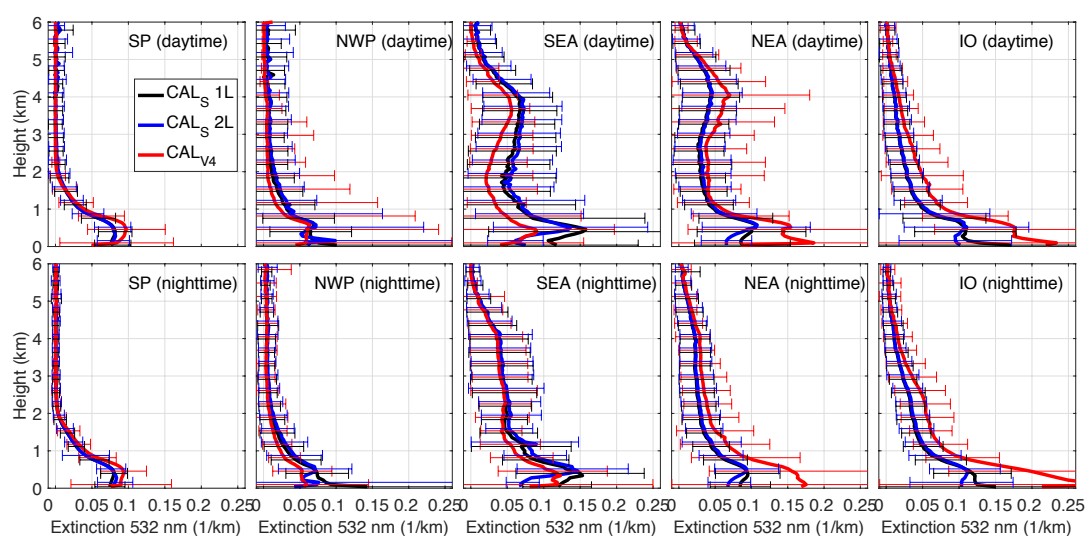

Figure 5: Mean aerosol extinction coefficient at 532 nm for the five regions defined in Fig. 4.
Upper and lower panels correspond to daytime and nighttime retrievals. CALIPSO-SODA
profiles are in black (1L) and blue (2L), and CALIPSO V4 is in red.

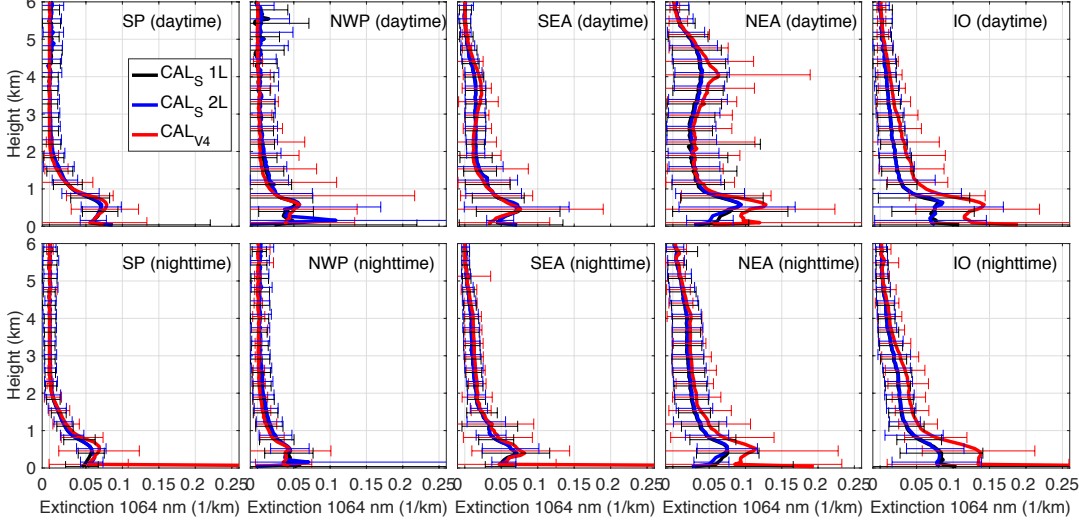

Figure 6: As in Figure 5 but for the 1064 nm channel.





Figure 7: Semi-annual daytime 532 nm lidar ratios. a) $LR_{1L}$ for spring-summer, b) $LR_{2L}$ for spring-summer, c) $LR_{1L}$ for autumn-winter, and d) $LR_{2L}$ for autumn-winter.

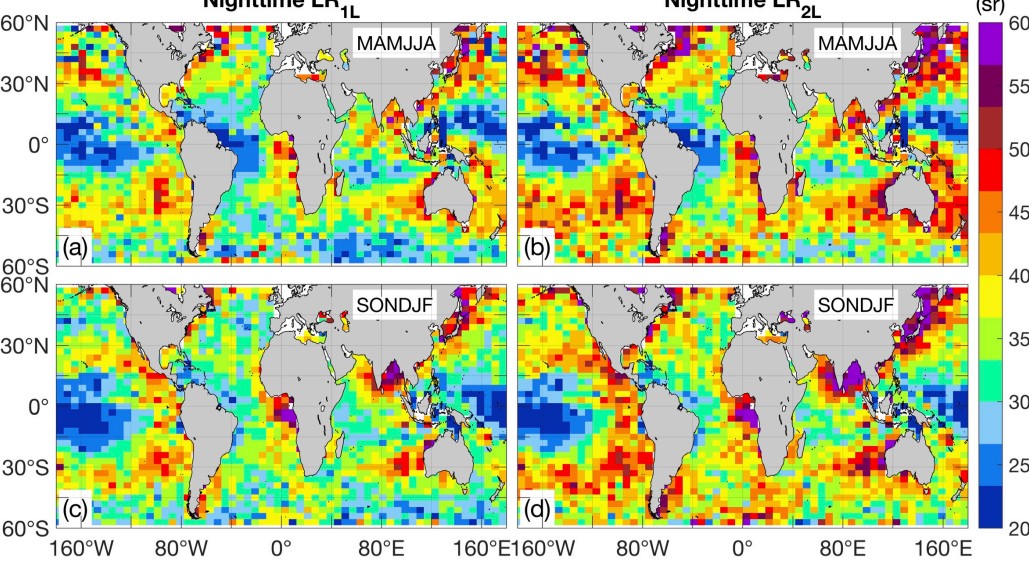

Figure 8: As in Figure 7 but for nighttime.





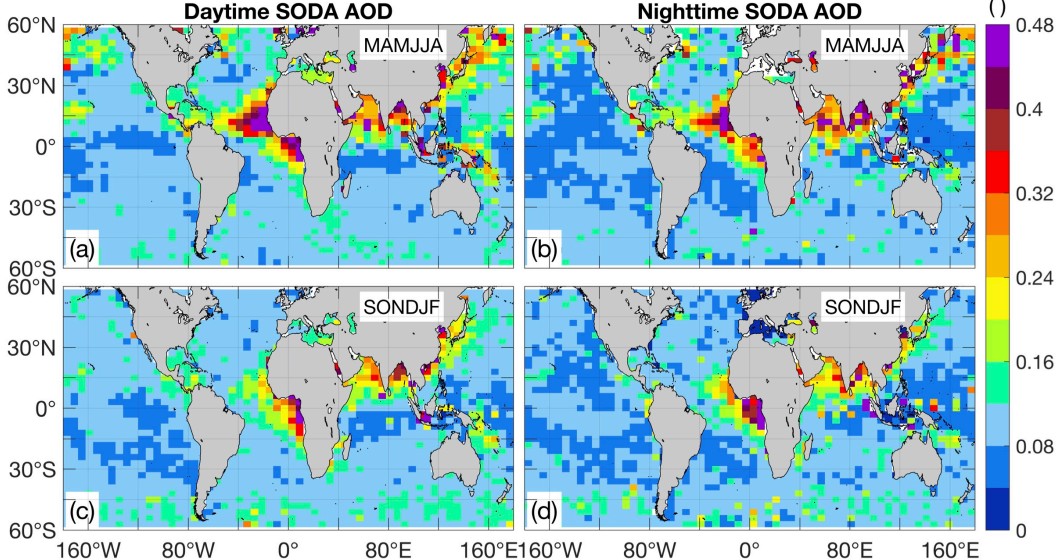

Figure 9: SODA AOD for daytime (a and c) and nighttime (b and d), spring-summer (MAMJJA) and autumn-winter (SONDJF).



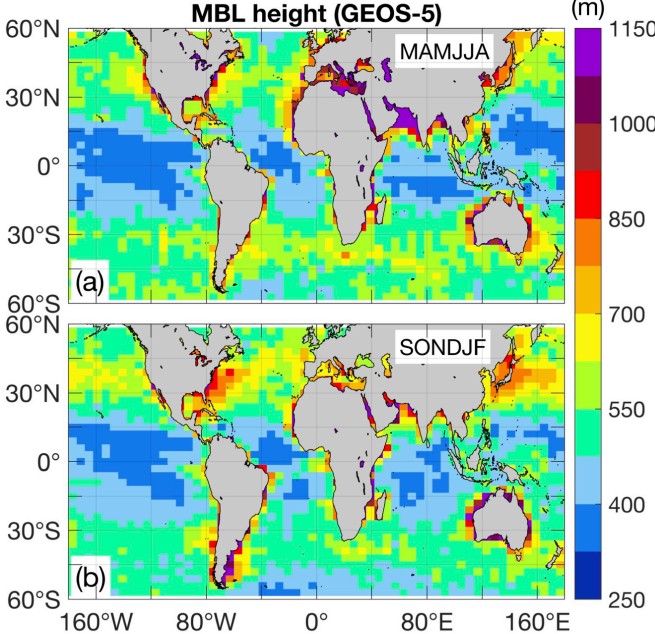

3     Figure 10: Daytime marine boundary layer height for a) spring-summer, and b) autumn-winter.





3   Figure 11: Daytime MBL 532 nm AOD based on 1L (left) and 2L (right).





1         Figure 12: Fraction of daytime AOD contributed by the marine boundary layer.

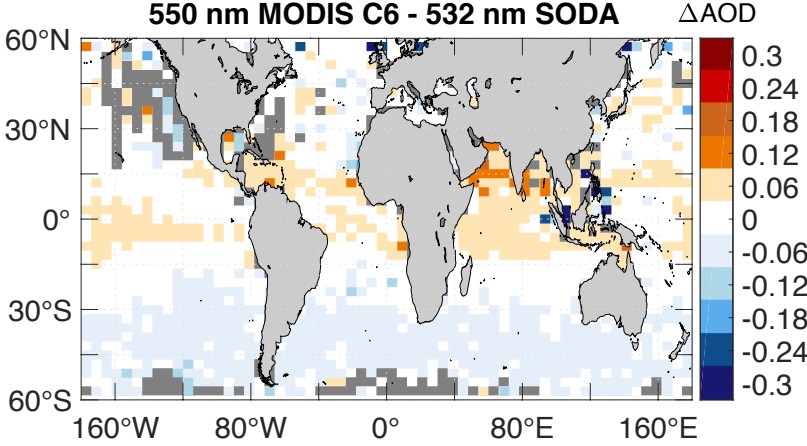

4   Figure 13: Mean AOD difference between matched 550 nm MODIS C6 and 532nm SODA

5                 daytime AOD for five months of 2010.



**Tables**
Table 1: Linear correlation coefficient (*r*), mean bias, and RSME between HSRL and SODA and
CALIOP Standard V4 AOD. Percentages are calculated relative to the mean HSRL AOD.

| CALIOP-based AOD | r | Mean bias | RSME |
|---|---|---|---|
| SODA | 0.96 | -0.024 (-17%) | 0.035 (24.2%) |
| Standard V4 | 0.94 | 0.014 (10%) | 0.044 (31.2%) |

Table 2: As in Table 1 but for CALIOP-SODA lidar ratio

| CALIOP SODA LR | r | Mean bias | RSME |
|---|---|---|---|
| 1 layer (1L) | 0.67 | -2.4 sr (-8.1%) | 7.4 sr (24.8%) |
| 2 layer (2L) | 0.74 | -3.9 sr (-12.5%) | 8.1 sr (26.0%) |

Table 3: As in Table 1 but for V4 and SODA aerosol extinction coefficient in the lower troposphere
(below 3.0 km).

| CALIOP-based extinction | r | Mean bias | RMSE |
|---|---|---|---|
| CALIOP V4 | 0.82 | 0.013 km$^{-1}$ (33.0%) | 0.043 km$^{-1}$ (106.0%) |
| SODA 1 layer (1L) | 0.78 | -0.0037 km$^{-1}$ (-9.2%) | 0.028 km$^{-1}$ (72.6%) |
| SODA 2 layer (2L) | 0.79 | -0.0029 km$^{-1}$ (-7.0%) | 0.028 km$^{-1}$ (73.8%) |

