# Peer review of "Novel aerosol extinction coefficients and lidar ratios over the ocean from"

_Atmospheric Measurement Techniques, 2018_

## Referee Comment (RC1) · Anonymous Referee #1 · 17 Dec 2018

The paper is well written. It contains original and interesting results, a nice technique is used by combing spaceborne lidar (CALIOP) and radar (CloudSat) observations. All this is highly appropriate to be published in AMT.

However, more comparisons with published (literature) observations of lidar ratios should be presented, and will improve the good paper.

My recommendation: Minor revisons.

Details:

P4, L10-12: Please be a bit more specific, more quantitative, if the extinction coefficient

is below 25 Mm-1 or the AOT is below 0.05 CALIOP will no detect this aerosol? Please, provide some kind of threshold numbers.

P6, L6-8: Is the two-layer method not similar to the approach of Ansmann, Appl. Opt, 45, 2006 (Ground-truth aerosol lidar observations: can the Klett solutions obtained from ground and space be equal for the same aerosol case?). Should probably be mentioned.

P6, L12-16: Regarding true marine lidar ratios, you may check and give reference to the papers of Gross et al., Tellus, 2011 (Cabo Verde, SAMUM2), ACP 2015 (Barbados, SALTRACE), Haarig et al., ACP, 2017 (SALTRACE, Barbados, wet and dry sea salt lidar ratios).

P7, L3: Great design of the campaign is visible in Fig.1! Well planned!

P7, L16: What is the truth? HSRL? How do you know, what the true AOD is?

P7, L27-29: column lidar ratio.... also given in Ansmann, Appl, Opt, 45, 2006.

P8, L1-8: Please explain better, first: 1L approach: all details, .... afterwards 2L approach, i.e., explain 1L and 2L separately, one after another. At the moment, too many details are given at the same time..., it took me some time to 'disentangle' the information properly.

P8, L14-19: Overestimation..., is that caused by the use of the Klett forward integration method? Could be mentioned...

P10, L18-21: Here, more comparisons with literature lidar ratio values would be good: Franke et al., GRL 2001, JGR 2003 (Indian Ocean, INDOEX, Maldives, Indian pollution aerosol, 2L structures...), Gross et al., , Tesche et al., both in Tellus 2011(eastern Atlantic, SAMUM2, Cabo Verde, summer and also winter, Tellus, 2011), Gross et al., 2015, Haarig et al. 2017, both in ACP (Caribbean, SALTRACE, Barbados, dust lidar ratios), Bohlmann et al, ACP, 2018, Polarstern cruises from the North to the South Atlantic, with Raman lidar aboard, also Kanitz et al., JGR 2013, issue 6. . And please

check also ... all the papers from Japanese, Chinese, and Korean groups. A good starting point may be the following recent paper in ACP: Vertical variation of optical properties of mixed Asian dust/pollution plumes according to pathway of air mass transport over East Asia S.-K. Shin, D. Müller, C. Lee, K. H. Lee, D. Shin, Y. J. Kim, and Y. M. Noh Atmos. Chem. Phys., 15, 6707-6720, https://doi.org/10.5194/acp-15-6707-2015, 2015.

Please check the reference list in this paper for more lidar ratio papers.

P12, L10: limited number of observations of lidar ratios. ... As mentioned please check the available literature..., and then 'update' this statement a little bit.

P12, L25 to P13, L30... and one has always to be careful with column lidar ratios, when marine particles are involved (so in the case of the SODA approach). The lidar ratio of sea salt is partly below 20sr, So these particles are rather efficient in backscattering of laser photons. As a consequence, their weight in the backscatter-weighted column integration... controls or can dominate the result...

Again, discuss the literature values (P13, L6, Kanitz, Bohlmann), L11-12, Franke et al., L15-16, Franke et al., L20-21, Haarig et al., Bohlmann et al.

All in all: An excellent paper!

---

## Referee Comment (RC2) · Anonymous Referee #2 · 3 Jan 2019

The authors describe a method for deriving aerosol lidar ratios over the oceans from combining CALIPSO and CloudSat data and present their findings for a year of data. The paper is of interest to the readership of AMT and well within the scope of the journal. I recommend publication only after major revision according to the points listed below.

Major issues

- The description of the methodologies used in this study could be improved so that it will be easier for the reader to follow.

[Figure]

- It would be good to have separate subsection for the different methods (1L and 2L).

- A flowchart of the procedure would also be helpful.

- Also, some details should be added in the description of the methodology. For instance:

  - Have any CAD scores been used to select the CALIOP data used for comparison? Or have you used any other quality-assurance critera for filtering CALIOP data (apart from the requirement for cloud-free conditions)?

  - Have you checked if GEOS-5 MBL heights agree with the layer in which the CALIPSO vertical feature mask shows marine aerosols?

  - Why are CALIOP and HSRL observations averaged over 0.5 degree in latitude?

  - Why is the upper boundary in Eq. (4) fixed at 6 km height? Have you checked that aerosol layers always extend to this height?

  - Why do you use five month in Section 5.1 and 12 months in Section 5.2?

  - Will there be a difference if cloud screening is performed before averaging the data?

- I find it really hard to assess the quality of the findings without any information on the number of matches for the data presented in Figures 4 to 13. For all I know, results could be based on 10 or 100000 data points per grid cell. Please add some maps that show the number of matches.

- I am surprised that the maps in Figures 7 and 8 show no clearer structure according to known aerosol transport pathways. For instance, I would expect a clear increase in LR (at least for the 2L method) for the transport of Saharan dust from Africa to South America during summer (particularly as this is visible in the

AOD plots in Figure 9). It rather seems that patches of increased LR occur in regions of persistent low-level cloud decks to the west of the continents. Information on the number of data points per grid cell is needed to assess the quality of the results.

- The difference between AOD from MODIS and SODA is quite large over some regions. This is discussed much too late in the text. It should also be explored what this means for the LR retrieval.

- The choice of reference literature is rather one-sided. The discussion mostly refers to Burton et al. (2012, 2013) and the authors seem to neglect a host of publications on the lidar ratio of different aerosol types and aerosol mixtures. In general, the discussion of the findings needs improvements in light of my earlier comments.

Minor issues

- Please make certain that acronyms are properly introduced when first mentioned

- Please connect statements such as "good agreement" to something more quantitative

- P3L6: omit one coefficient

- P3L6: Have you been varying the background values as well? What values have been used?

- P4L25: A systematic error of 0.059 for daytime AOD over the ocean seems very high to me, i.e. within the range of total AOD for clean marine conditions. Has this large systematic error in any way been considered in this study?

- P4L31: uncertainty calibrations = calibration uncertainties?

- P6L1: Should it not be $\sigma_a(z)$ here? $\sigma_m$ is known once you know $\beta_m$ as $LR_m = 8\pi/3$.

- P6L5: What's the effect of the SODA bias mentioned earlier to the LR retrieval?

- P9L22/23: Omit the description of line colours. This belongs into the figure caption.

- P12L26: Is the reference to Figure 9 correct?

- P13L4: Should be Figure 7a?

- Figure 1: Please note the meaning of the coloured circles in Figure 1a.

- Figures 5 and 6 could be moved to the supplement.

- Figure 14: What do the grey regions refer to?

---

## Author Comment (AC1) · 11 Mar 2019

**Reply to Referee #1**

We appreciate the reviewer's encouraging comments and valuable suggestions. Moreover, the bibliographical list provided by the reviewer was particularly relevant for interpreting our results in a more comprehensive context. Our responses to the reviewer's comments are addressed below (in blue):

Details:
P4, L10-12: Please be a bit more specific, more quantitative, if the extinction coefficient is below 25 Mm-1 or the AOT is below 0.05 CALIOP will no detect this aerosol? Please, provide some kind of threshold numbers.

The reviewer's request is addressed in the following sentence:

"Typically, the detection threshold is range-dependent, and varies as a function of molecular density, solar background and other instrument noise, and signal averaging (Vaughan et al., 2009). In terms of AOD, global analysis of CALIOP V3 daytime data by Toth et al. (2018) show that the "aerosol-free" columns reported by the CALIOP algorithm correspond to a mean MODIS AOD of 0.03-0.05. A similar analysis by Kim et al. (2017) shows that, as expected, CALIPSO extinction and AOD retrieval capabilities are substantially better at night than during the day. These authors estimate a maximum mean undetected extinction coefficient of ~0.006 $km^{-1}$ during daytime versus ~0.003 $km^{-1}$ at night (see their Fig. 5c)."

P6, L6-8: Is the two-layer method not similar to the approach of Ansmann, Appl. Opt, 45, 2006 (Ground-truth aerosol lidar observations: can the Klett solutions obtained from ground and space be equal for the same aerosol case?). Should probably be mentioned.

Ansmann (2006) is now properly acknowledged in our revised manuscript.

P6, L12-16: Regarding true marine lidar ratios, you may check and give reference to the papers of Gross et al., Tellus, 2011 (Cabo Verde, SAMUM2), ACP 2015 (Barbados, SALTRACE), Haarig et al., ACP, 2017 (SALTRACE, Barbados, wet and dry sea salt lidar ratios).

We appreciate the detailed list of references. The following lidar-related articles are included in the revised manuscript (more details in page 4):
Ansmann, A.: Ground-truth aerosol lidar observations: can the Klett solutions obtained from ground and space be equal for the same aerosol case?, *Appl. Optics*, 45, 3367–3371, 2006.
Groß, S., Freudenthaler, V., Schepanski, K., Toledano, C., Schäfler, A., Ansmann, A., and Weinzierl, B.: Optical properties of long-range transported Saharan dust over Barbados as measured by dual-wavelength depolarization Raman lidar measurements, *Atmos. Chem. Phys.*, 15, 11067-11080, https://doi.org/10.5194/acp-15-11067-2015, 2015.

Mona, L., Amodeo, A., Pandolfi, M., and Pappalardo, G.: Saharan dust intrusions in the Mediterranean area: Three years of Raman lidar measurements. *J. Geophys. Res.*, 111, D16203, doi:10.1029/2005JD006569, 2006.

Noh, Y. M., Kima, Y. J., and Muʿller, D.: Seasonal characteristics of lidar ratios measured with a Raman lidar at Gwangju, Korea in spring and autumn, *Atmos. Environ.*, 42, 2208–2224, doi:10.1016/j.atmosenv.2007.11.045, 2008.

Bohlmann, S., Baars, H., Radenz, M., Engelmann, R., and Macke, A.: Ship-borne aerosol profiling with lidar over the Atlantic Ocean: from pure marine conditions to complex dust–smoke mixtures, *Atmos. Chem. Phys.,* 18, 9661-9679, https://doi.org/10.5194/acp-18-9661-2018, 2018.

Tesche, M., Gross, S., Ansmann, A., Muʿller, D., Althausen, D., Freudenthaler, V., and Esselborn, M.: Profiling of Saha- ran dust and biomass-burning smoke with multiwavelength polarization Raman lidar at Cape Verde, *Tellus*, B63, 649–676, doi:10.1111/j.1600-0889.2011.00548.x, 2011.

P7, L3: Great design of the campaign is visible in Fig.1! Well planned!

The campaign was primarily intended for acquiring HSRL measurement to validate CALIOP data. We were fortunate of having these measurements for our study.

P7, L16: What is the truth? HSRL? How do you know, what the true AOD is? P7, L27-29: column lidar ratio. . .. also given in Ansmann, Appl, Opt, 45, 2006.

We added the following to explain that the HSRL is the truth in the context of our study:

" HSRL 532 nm AOD and aerosol extinction coefficient have been regularly validated against other airborne instruments, with biases less than 6% and 3%, respectively (Rogers et al. 2009), and generally to within 0.03 in comparison with AERONET AOD (Sawamura et al., 2017). The AOD product from the HSRL instrument makes use of the molecular channel which is a direct observation of atmospheric attenuation between the aircraft and the surface when compared against the GEOS-5 molecular density profile (Rogers et al. 2009). Since this method requires no assumptions about the lidar ratio or assumptions that the lidar ratio is constant, it provides a useful truth measurement in the context of this study."

P8, L1-8: Please explain better, first: 1L approach: all details, .... afterwards 2L approach, i.e., explain 1L and 2L separately, one after another. At the moment, too many details are given at the same time. . . , it took me some time to 'disentangle' the information properly.

We updated this section to read:

"The conventional method to solve eq. (3) follows Fernald (1984) and consists of iteratively solving for $\beta_a$, assuming a functional form of the lidar ratio LR(z). The LR selection is physically constrained by comparing the retrieved aerosol optical depth. ($AOD_{ret} = \int_0^z \sigma_a(z')dz'$) with SODA AOD ($AOD_{SODA}$), and LR is iteratively adjusted until the retrieved AOD matches the SODA AOD to within 0.001 or less (i.e., when $|AOD_{ret} - AOD_{SODA}| \leq 0.001$). While the shape of LR with height can be selected in different ways (e.g. Ansmann, 2006), here we opt for two assumptions, which in turn yield two independent sets of aerosol extinction and lidar ratio retrievals:

A. 1-layer lidar ratio (1LR): The simplest assumption is to consider one constant lidar ratio with height. This method is expected to perform well for atmospheric profiles characterized by only one aerosol type.

B. 2-layer lidar ratio (2LR): We also consider an additional scenario, which consists of treating the atmospheric column as two layers, that is, the marine atmospheric boundary layer (MBL) and a second aerosol layer of as-yet-undetermined composition. This method is intended to better capture specific events with two predominant aerosol types, particularly smoke over marine aerosols and dust over marine aerosols, which are particularly frequent over the Atlantic Ocean. The LR for the MBL is assumed constant at 25 sr, as suggested by HSRL measurements over the ocean (Burton et al., 2012; 2013). This lidar ratio is slightly higher than the one compiled by Kim et al. (2018, 23 sr). In contrast, 532 nm Raman lidar observations at Barbados encompass lidar ratios between 21 and 35 sr, with magnitudes primarily controlled by free tropospheric intrusion of dust (Groß et al., 2015). Similar range of MBL lidar ratio were observed in the eastern Atlantic by Bohlmann et al. (2018), with values modulated by the presence of dust-smoke aerosols. Without a-priori knowledge of MBL lidar ratio, the value prescribed here (25 sr) is within the range reported in previous studies over the ocean. $\sigma_a(z)$ and the upper layer LR are iteratively calculated using the Fernald method with the constraint provided by $AOD_{SODA}$, and LR =25 sr in MBL. MBL height is computed by applying the bulk Richardson number method (McGraw-Spangler and Molod, 2014). "

P8, L14-19: Overestimation. . ., is that caused by the use of the Klett forward integration method? Could be mentioned. . .

The aforementioned sentence is referring to the overestimation of CALIOP V4 aerosol extinction coefficient. The bias is primarily explained by the constant lidar ratio utilized by V4, which is higher than that observed during the field campaign.

P10, L18-21: Here, more comparisons with literature lidar ratio values would be good: Franke et al., GRL 2001, JGR 2003 (Indian Ocean, INDOEX, Maldives, Indian pollu- tion aerosol, 2L structures. . .), Gross et al., , Tesche et al., both in Tellus 2011(eastern Atlantic, SAMUM2, Cabo Verde, summer and also winter, Tellus, 2011), Gross et al., 2015, Haarig et al. 2017, both in ACP (Caribbean, SALTRACE, Barbados, dust lidar ratios), Bohlmann et al, ACP, 2018, Polarstern cruises from the North to the South At- lantic, with Raman lidar aboard, also Kanitz et al., JGR 2013, issue 6. . And please check also . . . all the papers from Japanese, Chinese, and Korean groups. A good starting point may be the following recent paper in ACP: Vertical

variation of optical properties of mixed Asian dust/pollution plumes according to pathway of air mass trans- port over East Asia S.-K. Shin, D. Müller, C. Lee, K. H. Lee, D. Shin, Y. J. Kim, and Y. M. Noh Atmos. Chem. Phys., 15, 6707-6720, https://doi.org/10.5194/acp-15-6707-2015, 2015.

Please check the reference list in this paper for more lidar ratio papers.

P12, L10: limited number of observations of lidar ratios. . .. As mentioned please check the available literature. . ., and then 'update' this statement a little bit.  Again, discuss the literature values (P13, L6, Kanitz, Bohlmann), L11-12, Franke et al., L15-16, Franke et al., L20-21, Haarig et al., Bohlmann et al.

We have included some of the suggested references in the manuscript. The inclusion of new references is primarily reflected in the revised discussion:

"As different aerosol types can be, to some extent, characterized by their lidar ratio, the reliability of CALIOP-SODA LR retrievals is qualitatively assessed by analyzing the consistency between the CALIOP-SODA LR spatial pattern and the regional occurrence of aerosol types as well as lidar measurements from several field campaigns over the ocean. Burton et al. (2012), using HSRL measurements over North America and the adjacent Atlantic Ocean, provide the following lidar ratios for a number of aerosol types: the highest LR (45-80 sr) are typically attributed to smoke and urban aerosols, LR of 25-50 sr and 40 sr are associated with dust and polluted maritime aerosols (respectively), and maritime aerosols are characterized by lidar ratios of less than 30 sr. For simplicity, we will primarily interpret daytime $LR_{1L}$ in Figures 9a and c for the following regions of interest:

[revised manuscript text omitted]

P12, L25 to P13, L30. . . and one has always to be careful with column lidar ratios, when marine particles are involved (so in the case of the SODA approach). The lidar ratio of sea salt is partly below 20sr, So these particles are rather efficient in backscattering of laser photons. As a consequence, their weight in the backscatter-weighted column integration. . . controls or can dominate the result. . .

The reviewer raises an interesting point. This is the primary motivation of why we also retrieve CALIOP-SODA extinction using the the 2-layer method (2L), in which the lidar ratio is assumed constant at 25 sr in the boundary layer, and the one of the upper layer is estimated using the Fernald-Klett method.

All in all: An excellent paper!

We appreciate the reviewer's kind words.

---

## Author Comment (AC2) · 11 Mar 2019

**Reply to Referee #2**

We appreciate the reviewer's insightful comments and valuable suggestions. His/her careful inspection of our manuscript gave us the opportunity to improve the quality of our work. The responses to the reviewer's comments are addressed below (in blue):

- 1. The description of the methodologies used in this study could be improved so that it will be easier for the reader to follow. – It would be good to have separate subsection for the different methods (1L and 2L). – A flowchart of the procedure would also be helpful.
  We followed Referee # 2's (and Referee # 1) recommendation concerning the description of 1L and 2L methods, which was modified to read:

"It follows that eq. (1) can be expressed in terms of LR and $\beta_m$ as:

$$\beta_{att}(Z) = \left(\beta_m(z) + \beta_a(z)\right) \cdot exp\left(-2\int_0^z \left(\sigma_m(z') + LR(z') \cdot \beta_a(z')\right)dz'\right) \ (3)$$

The conventional method to solve eq. (3) follows Fernald (1984) and consists of iteratively solving for $\beta_a$, assuming a functional form of the lidar ratio LR(z). The LR selection is physically constrained by comparing the retrieved aerosol optical depth. $(AOD_{ret} = \int_0^z \sigma_a(z')dz')$ with SODA AOD $(AOD_{SODA})$, and LR is iteratively adjusted until the retrieved AOD matches the SODA AOD to within 0.001 or less (i.e., when $|AOD_{ret} - AOD_{SODA}| \leq 0.001$). While the shape of LR with height can be selected in different ways (e.g. Ansmann, 2006), here we opt for two assumptions, which in turn yield two independent sets of aerosol extinction and lidar ratio retrievals:

A. 1-layer lidar ratio (1LR): The simplest assumption is to consider one constant lidar ratio with height. This method is expected to perform well for atmospheric profiles characterized by only one aerosol type.
B. 2-layer lidar ratio (2LR): We also consider an additional scenario, which consists of treating the atmospheric column as two layers, that is, the marine atmospheric boundary layer (MBL) and a second aerosol layer of as-yet-undetermined composition. This method is intended to better capture specific events with two predominant aerosol types, particularly smoke over marine aerosols and dust over marine aerosols, which are particularly frequent over the Atlantic Ocean.".

- 2. Have any CAD scores been used to select the CALIOP data used for comparison? Or have you used any other quality-assurance critera for filtering CALIOP data (apart from the requirement for cloud-free conditions)?
  In our original submission, we did not apply additional screening criteria to CALIOP V4. We have revised our results to only include CALIOP V4 retrievals with absolute CAD score of more than 50.:

  "To reduce ambiguities in the CALIOP aerosol classification scheme, we restrict the analysis to samples with cloud aerosol discrimination (CAD) score higher than |50|, equivalent to at least medium confidence in the CALIOP layer classification (Young et al., 2018)."

- Have you checked if GEOS-5 MBL heights agree with the layer in which the CALIPSO vertical feature mask shows marine aerosols?
  This is an interesting suggestion that is left for future work. We are planning to carry out a study that analyzes CALIOP-SODA LR as a function of CALIOP V4 aerosol types and their assigned lidar ratio.

- 3. Why are CALIOP and HSRL observations averaged over 0.5 degree in latitude?
  This was a typo that remained from an earlier version. The observations were averaged to 0.2°, instead. This is explained in the following section:
  "Lastly, satellite and airborne observations are spatially averaged to a common 0.2° resolution (in latitude). It is worth noting that although the effective horizontal resolution of CALIOP V4 is 5km, in reality, larger spatial averaging of the lidar signal are required (20 or 80 km) for tenuous aerosol layers to increase the aerosol layer detectability in the CALIPSO aerosol classification scheme. Thus, the use of an 0.2° horizontal average for comparing airborne and satellite observations is adequate when considering possible spatial averaging of CALIOP V4."

- 4. Why is the upper boundary in Eq. (4) fixed at 6 km height? Have you checked that aerosol layers always extend to this height?
  We now extend the upper boundary to around 6.5 km, consistent with the highest altitude with valid aerosol extinction retrievals from the HSRL. Where there is no appreciable aerosol, there is little sensitivity to the choice of lidar ratio, so choosing the layer top higher than the actual layer top does not add error. Even though changes in the analysis are modest, we updated the manuscript accordingly.

- 5. Why do you use five month in Section 5.1 and 12 months in Section 5.2?
  We included 12 months of CALIOP-SODA statistics because we were interested in providing semi-annual global maps that could be interpreted in the context of seasonal changes in aerosol types. Because we were interested in highlighting similarities/differences between SODA, MODIS, and CALIOP V4 we chose a 5-month period in which satellite data show high AOD over the ocean. This helped to better identify regions in which the different datasets disagree. We could have included a full year of data so the seasonal bias could be reported. However, we wanted to keep the focus on the LR retrievals, which is the novel contribution of this manuscript.

  6. Will there be a difference if cloud screening is performed before averaging the data?
  We realized that in our original submission, the cloud screening method was not clearly explained. In the revised version we wrote:
  "We first averaged 1km CALIOP-SODA to the V4 Level 2 nominal resolution (5km) and only samples with 5-km cloud-free scenes are utilized. This is intended for minimizing the potential effect of overcast scenes in the retrievals and aerosol swelling near the cloud edges (Várnai and Marshak, 2011). Then, CALIOP-SODA and CALIOP V4 data were further reduced by averaging the retrievals to a common 25 km resolution."
  In other words, a cloud screening was performed before averaging the data to 25 km resolution. We realized that the additional screening of removing 25 km elements with cloud cover of more than 67% did not produce any meaningful change in the mean maps.

Because of this, in our revised manuscript we are only applying the cloud screening to the 5-km averaged data.

- 7. I find it really hard to assess the quality of the findings without any information on the number of matches for the data presented in Figures 4 to 13. For all I know, results could be based on 10 or 100000 data points per grid cell. Please add some maps that show the number of matches.

This is a valuable suggestion, we have added the following figure:

[Figure]

Figure 8: Number of 25-km CALIOP-SODA samples contained in each semiannual average: a) daytime MAMJJA, b) Nighttime MAMJJA, c) Daytime SONDJF, d) Nighttime SONDJF.

We also indicate with crosses the regions for the LR and AOD maps where less than 15% of the maximum observable number of samples contribute to the average.

- 8. I am surprised that the maps in Figures 7 and 8 show no clearer structure ac- cording to known aerosol transport pathways. For instance, I would expect a clear increase in LR (at least for the 2L method) for the transport of Saharan dust from Africa to South America during summer (particularly as this is visible in the AOD plots in Figure 9). It rather seems that patches of increased LR occur in regions of persistent low-level cloud decks to the west of the continents. Information on the number of data points per grid cell is needed to assess the quality of the results.

We agree with the reviewer that the lack of spatial structure in the lidar ratio map, which could be attributed to the transatlantic dust transport, is puzzling. However, such spatial pattern is difficult to evaluate, given the lack of lidar ratio observations along the dust transport pathways. We provide the following discussion concerning the aerosol transport over the Atlantic Ocean:

"6.5. Central Pacific and northern Atlantic: The regions with the lowest LR are located over the tropical Pacific Ocean, where AOD is the lowest (Figure 11). An unanticipated result is the absence of a zonal band across the Atlantic that could be attributed to the westward transport of Saharan dust across the Atlantic Ocean. Unfortunately, due to the lack of in-situ observations along the Saharan dust pathways, the assessment of SODA LR over this region is challenging. Raman lidar data over the eastern Atlantic (Cape Verde), off the coast of western Africa, in spring show dust and smoke in the free troposphere and boundary layer with a mean LR of 54 sr (Tesche et al., 2011), and a dust layer thickness of about 4 km. Over the

same region, SODA LR is 40 sr, which increases up to 45-50 sr when LR is estimated using the (Barbados, 13.14° N, 59.62° W) in summer reveal the presence of maritime aerosols and dust, with lidar ratios of less than 40 sr in the boundary layer, and pure dust aerosols generally confined to the free troposphere (Groß et al., 2015). This suggests that the relatively low CALIOP-SODA LR over the Altantic basin may be explained by the contribution of maritime aerosols in the boundary layer. A more quantitative assessment, which includes the analysis of specific dust events, is left for future work."

Regarding the high lidar ratio over regions dominated by stratiform clouds, this could be explained by offshore transport of pollution (e.g. northeast Pacific). However, for more pristine environments such as the Southern Ocean, other environmental factors might be playing a role: "It is nevertheless surprising the high SODA lidar ratios retrieved over rather pristine regions, especially over the Southern Ocean, where maritime aerosols are expected to be the dominant aerosol type. A plausible factor that may help reconcile high LR for maritime aerosols is a lidar ratio increase with relative humidity (Ackerman 1998). Relative humidity could also explain the presence of LR >30 sr over stratocumulus cloud regimes, where high relative humidity is confined in the boundary layer."

9. The difference between AOD from MODIS and SODA is quite large over some regions. This is discussed much too late in the text. It should also be explored what this means for the LR retrieval.

We have moved the MODIS-SODA figure to the section where the differences between SODA and V4 AOD are presented. We also added an analysis of the sensitivity of the lidar ratio retrievals to uncertainties in AOD:

"However, errors in SODA AOD are plausible, especially when considering the sometimes large differences between SODA and MODIS AOD (>0.06, Figure 5). To assess the uncertainty in the retrieved CALIOP-SODA LR attributed to errors in SODA AOD, we assume a ±20% perturbation in SODA AOD and estimated LR. A 20 % AOD error is similar to the 24 % RMSE between SODA and the airborne HSRL AOD (Section 4). For one CALIPSO overpass we found that a 20% higher SODA AOD gives rise to a 5.4 sr increase in lidar ratio, or equivalent to a 14.4% lidar ratio change relative to the LR constrained with unperturbed AOD. Similarly, a 20% lower SODA AOD yields a 6.0 sr decrease in lidar ratio (-16.0%). These results are analogous to the $\Delta$AOD uncertainty of 18 % (for AOD=0.15) attributed to a 15% error in the lidar ratio prescribed by the CALIPSO algorithm, derived using the AOD error equation in Winker et al. (2009)."

- 10. The choice of reference literature is rather one-sided. The discussion mostly refers to Burton et al. (2012, 2013) and the authors seem to neglect a host of publications on the lidar ratio of different aerosol types and aerosol mixtures. In general, the discussion of the findings needs improvements in light of my earlier comments.

We now include the following lidar-related articles:

Ansmann, A.: Ground-truth aerosol lidar observations: can the Klett solutions obtained from ground and space be equal for the same aerosol case?, *Appl. Optics*, 45, 3367–3371, 2006.

Groß, S., Freudenthaler, V., Schepanski, K., Toledano, C., Schäfler, A., Ansmann, A., and Weinzierl, B.: Optical properties of long-range transported Saharan dust over Barbados as measured by dual-wavelength depolarization Raman lidar measurements, *Atmos. Chem. Phys.*, 15, 11067-11080, https://doi.org/10.5194/acp-15-11067-2015, 2015.

Mona, L., Amodeo, A., Pandolfi, M., and Pappalardo, G.: Saharan dust intrusions in the Mediterranean area: Three years of Raman lidar measurements. *J. Geophys. Res.*, 111, D16203, doi:10.1029/2005JD006569, 2006.

Noh, Y. M., Kima, Y. J., and Muʿller, D.: Seasonal characteristics of lidar ratios measured with a Raman lidar at Gwangju, Korea in spring and autumn, *Atmos. Environ.*, 42, 2208–2224, doi:10.1016/j.atmosenv.2007.11.045, 2008.

Bohlmann, S., Baars, H., Radenz, M., Engelmann, R., and Macke, A.: Ship-borne aerosol profiling with lidar over the Atlantic Ocean: from pure marine conditions to complex dust–smoke mixtures, *Atmos. Chem. Phys.,* 18, 9661-9679, https://doi.org/10.5194/acp-18-9661-2018, 2018.

Tesche, M., Gross, S., Ansmann, A., Muʿller, D., Althausen, D., Freudenthaler, V., and Esselborn, M.: Profiling of Saha- ran dust and biomass-burning smoke with multiwavelength polarization Raman lidar at Cape Verde, *Tellus*, B63, 649–676, doi:10.1111/j.1600-0889.2011.00548.x, 2011.

- Please make certain that acronyms are properly introduced when first mentioned
  Point taken, thanks.
- Please connect statements such as "good agreement" to something more quantitative
  In the revised version, we made sure to include quantitative information (bias, correlation coeff., etc) when compared different datasets.

- P3L6: omit one coefficient
  We would appreciate further clarification from the reviewer, as we could not find the sentence the reviewer was referring to.

- P3L6: Have you been varying the background values as well? What values have been used?
  We would appreciate further clarification from the reviewer.

- P4L25: A systematic error of 0.059 for daytime AOD over the ocean seems very high to me, i.e. within the range of total AOD for clean marine conditions. Has this large systematic error in any way been considered in this study?
  See our response to comment #9.

- P4L31: uncertainty calibrations = calibration uncertainties?
  Corrected, thanks
  P6L1: Should it not be $\sigma_a(z)$ here? $\sigma_m$ is known once you know $\beta_m$ as $LR_m = 8\pi/3$.
  The reviewer is correct. The text has been modified accordingly
- P6L5: What's the effect of the SODA bias mentioned earlier to the LR retrieval?
- See our response to comment #9.

- P9L22/23: Omit the description of line colours. This belongs into the figure caption.
- Done

- P12L26: Is the reference to Figure 9 correct?
  Corrected, thanks

- P13L4: Should be Figure 7a?
  The discussion section was modified and the paragraph was slightly changed.

- Figure 1: Please note the meaning of the coloured circles in Figure 1a.
  The colored circles indicate individual flights used in this study.

- Figures 5 and 6 could be moved to the supplement.
  We prefer to leave the figures in the manuscript as we feel that Figs. 5 and 6 provide a good example of how differences in the selection of lidar ratio can propagate to changes in aerosol extinction coefficient.

- Figure 14: What do the grey regions refer to?
  Gray regions refer to grids with no available MODIS-SODA datasets. The caption was modified accordingly.

---

## Author Response (AR2)

**Response to Reviewer 2**

Once again, we appreciate Reviewer2's suggestion of double checking that the figures are properly referenced. We made the following changes:

1. Caption for Figure 7 was corrected to read: "Figure 7: As in Figure 6 but for the 1064 nm channel."
2. Page 11, Lines 24: The correct figure to be referenced is Fig. 6 (instead of 5).
3. The reviewer is correct, the coauthor should be SB instead of RB (author contribution).